# Korean Red Ginseng and Ginsenoside Rg3 Suppress Asian Sand Dust-Induced Epithelial–Mesenchymal Transition in Nasal Epithelial Cells

**DOI:** 10.3390/molecules27092642

**Published:** 2022-04-20

**Authors:** Seung-Heon Shin, Mi-Kyung Ye, Dong-Won Lee, Mi-Hyun Chae, You-Jin Hwang

**Affiliations:** 1Department of Otolaryngology-Head and Neck Surgery, School of Medicine, Catholic University of Daegu, Daegu 42472, Korea; miky@cu.ac.kr (M.-K.Y.); neck@cu.ac.kr (D.-W.L.); leonen@hanmail.net (M.-H.C.); 2Department Biomedical Engineering, College of Health Science, Gachon University, Incheon 21936, Korea; gene@gachon.ac.kr

**Keywords:** Korean red ginseng, ginsenoside Rg3, nasal epithelial cell, Asian sand dust, epithelial–mesenchymal transition

## Abstract

Chronic rhinosinusitis (CRS) is characterized by chronic inflammation of the sinonasal mucosa with epithelial dedifferentiation toward the mesenchymal phenotype, known as the epithelial–mesenchymal transition (EMT). Asian sand dust (ASD) can induce nasal mucosal inflammation and cause the development of EMT. Korean red ginseng (KRG) and ginsenoside Rg3 have been used as traditional herbal medicines to treat various diseases. The aim of this study was to investigate their effect on ASD-induced EMT in nasal epithelial cells. Primary nasal epithelial cells were incubated with ASD with or without KRG or Rg3, and the production of transforming growth factor-β1 (TGF-β1) and interleukin (IL)-8 was measured. EMT markers were determined by RT-PCR, Western blot analysis, and confocal microscopy, and transcription factor expression by Western blot analysis. The effect on cell migration was evaluated using the wound scratch assay. Results showed ASD-induced TGF-β1 production, downregulation of E-cadherin, and upregulation of fibronectin in nasal epithelial cells. KRG and Rg3 suppressed TGF-β1 production (31.7% to 43.1%), upregulated the expression of E-cadherin (26.4% to 88.3% in mRNA), and downregulated that of fibronectin (14.2% to 46.2% in mRNA and 52.3% to 70.2% in protein). In addition, they suppressed the ASD-induced phosphorylation of ERK, p38, and mTOR, as well as inhibiting the ASD-induced migration of nasal epithelial cells (25.2% to 41.5%). The results of this study demonstrate that KRG and Rg3 inhibit ASD-induced EMT by suppressing the activation of ERK, p38, and mTOR signaling pathways in nasal epithelial cells.

## 1. Introduction

Epithelial–mesenchymal transition (EMT) is a reversible biologic process in which epithelial cells change their biochemical properties into mesenchymal-like cells with in-creased migration capacity, invasiveness, and production of the extracellular matrix [1,2]. EMT induced by chronic stress, various types of inflammation, and malignant progression is associated with tissue remodeling, wound healing, and fibrosis [3,4]. EMT can be divided based on three different biologic settings: type 1 occurs during implantation, embryogenesis, and organ development; type 2 is associated with tissue regeneration and organ fibrosis; and type 3 takes place during cancer progression and metastasis [3]. During the EMT process, epithelial markers, such as E-cadherin, are downregulated, while mesenchymal markers, such as N-cadherin, vimentin, and fibronectin, are upregulated [1,3].

Chronic rhinosinusitis (CRS) is a chronic inflammatory disease of the nasal and sinus mucosa characterized by inflammatory cell infiltration, tissue edema, epithelial dysfunction, and epithelial hyperplasia [5]. Airway remodeling in CRS includes the increased accumulation of extracellular matrix (ECM), subepithelial edema, and basement membrane thickening. Tissue remodeling with EMT is commonly found in CRS, where epithelial cells differentiate toward a more mesenchymal phenotype, with a loss of E-cadherin and increase in various mesenchymal markers [5,6]. The transforming growth factor (TGF)-β, mitogen-activated protein kinase (MAPK), small mother against decapentaplegic (SMAD) proteins, and the Snail/Slug signaling pathway are involved in the development and progression of EMT in CRS [6,7,8].

Asian sand dust (ASD) originates from the Gobi and Mongol deserts and is trans-ported over East China and across the North Pacific Ocean, reaching the coasts of the United States [9]. Inhaled ASD is directly in contact with respiratory epithelial cells and it induces the production of chemical mediators with inflammatory cell infiltration [10]. ASD causes the deterioration of the innate defense system of the respiratory mucosa with exacerbation of airway inflammatory and allergic diseases. In addition, ASD contains various chemical compounds, microbial agents, and different sizes of particulate matter (PM). The latter induces epithelial barrier dysfunction, which causes increased cell permeability, upregulation of extracellular matrix (ECM) accumulation, and loss of epithelial polarity and junctional proteins [11,12]. PM leads to the pathogenesis of CRS and nasal polyps through the formation of EMT in the nasal mucosa [11].

Korean Red ginseng (KRG; heat-processed Panax ginseng Meyer) is commonly used as an oriental medicine to treat allergic, inflammatory, and cancerous conditions [13,14,15]. KRG and ginsenosides, the active constituents of KRG, have been shown to inhibit cancer cell metastasis and cancer development, and to ameliorate liver fibrosis through the inhibition of EMT [14,16,17]. ASD and its PM induce chronic airway inflammatory diseases, and chronic stimulation by PM can cause the development of EMT and fibrosis of the airway mucosa [18]. The aim of this study was to investigate whether EMT can be induced by stimulation with ASD, and to evaluate the effect of KRG and ginsenoside Rg3 on ASD-induced EMT in nasal epithelial cells.

## 2. Results

### 2.1. Cytotoxicity of ASD, KRG, and Ginsenoside Rg3 in Nasal Epithelial Cells

To determine the cytotoxic effect of ASD, KRG, and Rg3, nasal epithelial cells were treated with various concentrations of ASD, KRG, and Rg3 for 24, 48, and 72 h. The established viability of epithelial cells was examined using the MTT assay to generate a cell titration curve. Cell viability significantly decreased at Rg3 and ASD concentrations of 100 μg/mL after 24 h of incubation, while, at KRG concentrations less than 500 μg/mL, the survival of nasal epithelial cells was not affected. When the cells were treated with 50 μg/mL of ASD and 500 μg/mL of KRG or 50 μg/mL of Rg3, their survival was not significantly affected after 72 h (Figure 1). Therefore, 50 μg/mL of ASD, less than 500 μg/mL of KRG, and 50 μg/mL of Rg3 were used for further experiments.

### 2.2. Effects of KRG and Rg3 on ASD-Induced Production of IL-8 and TGF-β1

The treatment of nasal epithelial cells with ASD led to a significant increase in IL-8 and TGF-β1 production. Pretreatment with 50 and 100 μg/mL of KRG and 10 μg/mL of Rg3 significantly inhibited TGF-β1 production (31.7% to 43.1%). However, ASD-induced IL-8 production was not suppressed during KRG or Rg3 treatments (Table 1).

### 2.3. Effects of KRG and Rg3 on ASD-Induced EMT Marker mRNA and Protein Expression

Among several EMT markers, the present study evaluated E-cadherin, fibronectin, and vimentin mRNA and protein expression. It was shown that ASD downregulated E-cadherin mRNA expression, and that KRG and Rg3 inhibited this ASD-induced down-regulation. ASD upregulated fibronectin mRNA and protein from nasal epithelial cells. These upregulations were suppressed by KRG or Rg3 treatments. However, ASD did not influence the expression of E-cadherin protein, vimentin mRNA, and vimentin protein from nasal epithelial cells (Figure 2).

To determine the cellular expression of E-cadherin and fibronectin, an immunofluorescence analysis was performed. Nasal epithelial cells treated with ASD had a lower florescence intensity for E-cadherin and a higher one for fibronectin compared to untreated cells. KRG and Rg3 inhibited ASD-induced E-cadherin and fibronectin expression in nasal epithelial cells (Figure 3).

### 2.4. Effects of KRG and Rg3 on ASD-Induced Transcription Factor Expression

To investigate the effects of KRG and Rg3 on EMT-related transcription factors in nasal epithelial cells, the phosphorylation of the MAPK, PI3K/Akt, and mTOR components was determined through Western blot analysis. The ASD-induced upregulation of the phosphorylation of ERK, p38, and mTOR expression was significantly suppressed by Rg3, and KRG suppressed the phosphorylation of ERK and mTOR. However, ASD did not influence the phosphorylation of JNK, PI3K, and Akt in nasal epithelial cells (Figure 4).

### 2.5. Effects of KRG and Rg3 on ASD-Induced Migration of Nasal Epithelial Cells

The migration of nasal epithelial cells was determined by scratching adherent cells with a pipette tip, and then measuring the distance that cells migrated over a period of 48 h (Figure 5A). Compared to the control group, ASD-induced epithelial cell migration took place further away from the initial scratched area. Pretreatment with 50 μg/mL and 100 μg/mL of KRG and 10 μg/mL of Rg3 significantly inhibited the ASD-induced cell migration (Figure 5B). The expression of F-actin, an important component of the cytoskeleton that induces tissue remodeling and cell migration [19], was recognized via immunofluorescence analysis. KRG and Rg3 suppressed ASD-induced F-actin expression in nasal epithelial cells (Figure 5C).

## 3. Discussion

ASD originates from the Chinese and Mongolian deserts and is transported to North America across the North Pacific Ocean. The chemical and microbial components of ASD influence the development and exacerbation of allergic and infectious inflammatory air-way diseases [20,21]. PM < 10 μm (PM10) is the main component of ASD (53–71%), and is in direct contact with airway epithelial cells, inducing the production of inflammatory chemical mediators with inflammatory cell infiltration in the airway mucosa [10,22]. Tis-sue remodeling with EMT is typically observed in chronic inflammation and tissue injury. This epithelial dedifferentiation toward mesenchymal changes is commonly observed in CRS [5,6]. The exposure of bronchial epithelial cells to PM induces the development of EMT with decreased E-cadherin, increased mesenchymal markers, and impaired epithelial barrier function [22,23]. PM has been shown to induce EMT and the development of CRS in mice [11]. The present study demonstrated the inhibitory effects of KRG and Rg3 on EMT in primary nasal epithelial cells exposed to ASD. In particular, the results showed that ASD induced the repression of E-cadherin expression and induction of fibronectin in nasal epithelial cells. However, N-cadherin and vimentin expression was not influenced by stimulation with ASD. In this study, autoclaved ASD was used to eliminate the potential effects of microbial components contained in the dust; therefore, the ASD particles alone may not strongly induce EMT in nasal epithelial cells, compared to ASD or PM, which contain organic, inorganic, and microbial components. The ASD particles may have been different from those used in other studies, because the composition of the ASD mixture depends on the location where it is collected, the sources of emission near the collection site, and the season in which the dust is sampled.

KRG is commonly used in Asian traditional medicine because of its various biological and immunological properties. Ginseng possesses more than 150 different ginsenoside saponin types. Heat converts fresh ginseng into red ginseng and increases the concentration of the Rg2, Rg3, and Rh1 ginsenosides [24]. KRG produces anti-inflammatory effects through the inhibition of MAPK, NF-κB, and c-Fos activities, and Rg3 inhibits the activation of COX-2, inducible nitric oxide synthase, and the production of proinflammatory cytokines [24,25]. EMT can be induced by various growth factors and cytokines, such as TGF-β1 and the epidermal growth factor. In CRS, the TGF-β1 level is approximately 3.5 ng/mL in CRSwNP tissue, and a concentration of 5 to 10 ng/mL of this factor was used to induce EMT in nasal epithelial cells [6,26]. However, ASD was shown to produce 46.6 ± 4.7 pg/mL of TGF-β1 from nasal epithelial cells. This quantity was very low to induce EMT. It was then inferred that the EMT marker mRNA, E-cadherin and fibronectin, EMT marker protein, and fibronectin expression were influenced by ASD. Although it was not certain whether the TGF-β1 was produced by ASD-induced EMT in this study, it was shown that KRG and Rg3 inhibited the ASD-induced TGF-β1 production in nasal epithelial cells and these anti-inflammatory properties suppressed EMT. The inhibitory effects of KRG and Rg3 on fibronectin expression may suppress EMT in nasal epithelial cells. Fibronectin is an important mesenchymal protein that plays a significant role in tissue remodeling, wound healing, and fibrosis. The inhibition of fibronectin activity suppresses cell migration and EMT development [27].

In EMT, epithelial cells are typically found to lose their polarity, junctional proteins, and epithelial adhesion molecules. The immunologic mechanisms of EMT development in CRS are complex, and previous studies have shown that the immunoreactivity of TGF-β1, MAPK, Snail/Slug, and Smad2/3 was increased in the nasal mucosa of CRS patients [8,28]. A hypoxic status in CRS induces EMT through the hypoxia inducible factor-1 and Smad3 signaling pathway [28]. In CRSwNP, the advanced glycosylation end-products/receptors of advanced glycosylation end-products/ERK signaling pathway plays an important role in the development of EMT [8]. PM activates the EMT inducers and consequent phosphorylation of Smad2/3 [12]. Various immunologic mechanisms are suggested based on the types of stimulant or cells for EMT studies. KRG inhibits hypoxia-induced and TGF-β1-induced EMT by blocking the NF-κB and ERK 1/2 pathway, and the Smad/p38/ERK pathway, respectively, in malignant cell lines [14,17]. In this study, ASD induced the phosphorylation of ERK, p38, and mTOR, but not that of JNK, PI3K, and Akt, while KRG and Rg3 suppressed the phosphorylation of ERK, p38, and mTOR in nasal epithelial cells.

The pharmacokinetic properties of KRG or ginsenosides are not completely understood. The absorption rate of orally taken ginseng or ginsenosides is low and their metabolites and biological activity are affected by interaction with the gut microbiota, such as *Fusobacterium, Eubacterium, Bifidobacterium*, and *Rhodanobacter* [29,30]. Topical application of KRG or ginsenosides in the sinonasal mucosa to control local inflammation may interact with the sinonasal microbiota, such *as Corynebacterium sp.*, *Staphylococcus aureus*, *Pseudomonas aeruginosa*, and *Ptptoniphilus*, and could change their bioproperties. The active metabolites may disperse and be cleared from the sinonasal mucosa to the gut through mucociliary clearance. Even though active metabolites enter into the gut, only a small portion of metabolites may show bioavailability. Therefore, topical application of KRG or ginsenosides may be more efficient and safer for use in clinical situations. Further studies are needed to understand the bioavailability and pharmacokinetic mechanisms of topically applied ginseng compounds in the sinonasal mucosa.

There are some limitations in explaining the immunopharmacological phenomenon occurring in the sinonasal mucosa. We demonstrated that KRG and Rg3 can inhibit ASD-induced EMT. However, other ginsenosides, such as Rg4, Rg5, and Rb1, strongly suppress inflammation or EMT-associated transcription factors [31,32]. Thus, we need further studies, using other ginsenosides, to compare their EMT inhibition ability with that of Rg3. We used 50 μg/mL of ASD to induce EMT in nasal epithelial cells. However, the used concentration in this study did not represent natural air conditions as the mean PM10 level (40.7 μg/mL) and PM2.5 (22.3 μg/mL) level in 2019 based on the data from the Korea Meteorological Administration indicates. Moreover, ASD contains various chemical and microbial components. In this study, we used autoclaved ASD particles, but if we use non-treated ASD, it may more strongly induce EMT in the sinonasal mucosa.

## 4. Materials and Methods

### 4.1. Preparation of KRG and ASD

The standardized water extract of KRG and Rg3 was supplied by the KT&G Corporation (Daejeon, Korea). Panax ginseng Meyer was cultivated for 6 years in the Korean peninsula. The KRG derived from ginseng roots was manufactured by steaming and drying white ginseng. Fresh ginseng was steamed at 90–100 °C, and then dried at 50–80 °C. Red ginseng extracts were extracted repeatedly at 85 °C until sufficiently obtained. The hot water extract was prepared and provided by KT&G. Ginsenoside Rg3 content was determined by high-performance liquid chromatography. KRG contains approximately 50 kinds of ginsenosides; representative protopanaxtriol ginsenosides are Rg1—3.3 mg/g and Re—2.0 mg/g, and protopanaxadiol ginsenosides are Rb 1—5.8 mg/g, Rb2—2.3 mg/g, and Rc—1.7 mg/g. Analyses were performed using the general analytic method of ginsenosides of the Korea Food and Drug Administration [33]. The chemical structure of Rg3 is proposed in Figure 6.

ASD was collected using a high-volume air sampler (HV-500F, Shibat Scientific Technology, Saitama, Japan), during an ASD warning period in Incheon. After collection, the filter paper was washed with 10 mL of phosphate buffer solution. The fluid was filtered and the PM particles were collected and then centrifuged. The collected ASD was autoclaved at 121 °C for 15 min and stored at −20 °C. The chemical components of ASD are shown in Table 2.

### 4.2. Primary Nasal Epithelial Cell Culture and Cell Viability Assays

Primary nasal epithelial cells were isolated from the inferior turbinate of 12 subjects (six men and six women; average age 43.6 ± 12.7 years) during septal surgery procedures. Subjects were excluded if they had active inflammation, allergies, or if they had received antibiotics, antihistamines, or other medications for at least 4 weeks preoperatively. The allergy status was determined using the skin prick test.

Specimens were placed in Ham’s F-12 medium supplemented with 100 IU penicillin, 100 μg/mL streptomycin, and 2 μg/mL amphotericin B. The nasal mucosa was rinsed with Ham’s F-12 medium supplemented with antibiotics and was incubated with 0.1% dispase (Roche Diagnostics, Mannheim, Germany) for 16 h at 4 °C. The epithelial cells were isolated by gentle agitation and cell suspensions were filtered through a No. 60 mesh cell dissociation sieve. The cells were suspended in Ham’s F-12 medium supplemented with antibiotics, 150 μg/mL glutamine, 5 μg/mL transferrin, 25 ng/mL epithelial growth factor, 15 μg/mL endothelial cell growth supplement, insulin 5 IU/mL 200 pM triiodothyronin, 100 nM hydrocortisone, and 15% fetal calf serum. Cell suspensions at a concentration of 1 × 10^6^ cells/mL were grown to 80–90% confluence in culture plates placed in a 5% CO_2_ humidified incubator at 37 °C.

Nasal epithelial cells were incubated with ASD at concentrations of 0, 50, 100, 250, and 500 μg/mL for 72 h, to evaluate its cytotoxic effects. Cell cytotoxicity was determined using a CellTiter-96^®^ aqueous one solution cell proliferation assay kit (Promega, Madison, WI, USA). For this assay, tetrazolium compound and phenazine ethosulfate were added to each well and plates were incubated for 4 h at 37 °C in a 5% CO_2_ chamber. Color intensities were assessed using a microplate reader at wavelength of 490 nm. The optical density correlates with the metabolic activity and the number of viable cells [34]. The cytotoxic effects of KRG (0 to 500 μg/mL) and Rg3 (0 to 1000 μg/mL) were also determined using a CellTiter-96^®^ aqueous one solution cell proliferation assay kit (Promega).

### 4.3. Analysis of IL-8 and TGF-β1 Production from Nasal Epithelial Cells

Nasal epithelial cells were pretreated with or without 5, 50, and 100 μg/mL of KRG, or 10 μg/mL of Rg3 for 1 h, followed by stimulation with 50 μg/mL of ASD for 48 h. Interleukin (IL)-8 and transforming growth factor (TGF)-β1 levels in supernatants were quantified using the enzyme-linked immunosorbent assay (ELISA) kit (R&D Systems, Minneapolis, MN, USA). The absorbance at 450 nm was determined using an ELISA reader (BMG Labtech, Ortenaukreis, Germany).

### 4.4. Analysis of the Expression of EMT Marker mRNA and Protein

After 48 h of treatment, with ASD with or without pretreatment with KRG or Rg3 for 1 h, nasal epithelial cells were harvested. E-cadherin, fibronectin, and vimentin mRNA ex-pression was determined using real-time RT-PCR. The cells were placed in a cryo-tube, 1 mL of TRIzole reagent was added, and RNA was extracted according to the manufacturer’s instructions (Roche Diagnostics, Mannheim, Germany). RNA purity and concentration were measured using a spectrophotometer (Beckman, Mountain View, CA, USA). Using amplified cDNA, the PCR of E-cadherin, fibronectin, vimentin, and β-actin was performed with a SYBR green PCR core kit (PE Applied Biosystems, Foster, CA, USA) in a GeneAmp 5700 system (PE Applied Biosystems). The primer sequences used and amplified products were as follows: E-cadherin sense 5′-ATA GAG AAC GCA TTG CCA CAT ACA-3′ and antisense 5′-TTC TGA TCG GTT ACC GTG ATC A-3′ (130 bp), fibronectin sense 5′-GCC AGA TGA TGA GCT GCA C-3′ and antisense 5′-GAG CAA ATG GCA CCG AGA TA-3′ (142 bp), and vimentin sense 5′-AAC CTG GCC GAG GAC ATC A-3′ and antisense 5′-TCA AGG TCA AGA CGT GCC AGA-3′ (134 bp). The annealing temperature was 60 °C. All samples were amplified in triplicate. mRNA expression levels were normalized to the median value of the endogenous control, β-actin. The relative mRNA levels were qualified using the relative quantification 2 ΔΔCT method.

E-cadherin, fibronectin, and vimentin protein levels were determined using Western blot analysis. After 1 h of exposure to ASD and KRG or Rg3, nasal epithelial cells were harvested and lysed in an ice-cold RIPA buffer (Thermo Fisher Scientific, Rockford, IL USA). Cell lysates were collected and subjected to NuPAGE 4–12% Bis-Tris gel electrophoresis (Thermo Fisher Scientific); then, they were transferred onto nitrocellulose mem-branes (Bio-Rad, Berkeley, CA, USA). These were blocked with a membrane blocking solution (Thermo Fisher Scientific), and were incubated with antibodies against E-cadherin, fibronectin, vimentin, and GAPDH (Santa Cruz Biotechnology, Santa Cruz, CA, USA). After 24 h of incubation, the membranes were washed and were then treated with peroxidase-conjugated anti-mouse immunoglobulin G (Santa Cruz Biotechnology). Bands were visualized using the SuperSignal West Pico Chemiluminescent Substrate (Pierce, Rockford, IL, USA) and Gel Doc XR+ (Bio-Rad). Band densities were measured using the multi Gauge v.2.02 software (Fujifilm, Tokyo, Japan) and were expressed as a percentage of treated versus untreated cells.

### 4.5. Immunofluorescence Study of EMT Marker and F-Actin

Immunofluorescence assays were carried out as previously described [35]. Serum-starved cells were pretreated with or without KRG or Rg3 for 1 h, followed by stimulation with ASD for 48 h. They were then fixed with 4% paraformaldehyde for 15 min at room temperature and were washed three times. Subsequently, the cells were incubated with E-cadherin (BD transduction Laboratories, Franklin Lake, NJ, USA) and fibronectin (Santa Cruz biotechnology) primary antibodies overnight at 4 °C. FITC-conjugated anti-rabbit Ig G (1:100), 2 h at room temperature, was used for secondary incubation. Nuclei were counterstained with DAPI reagent. For F-actin staining, cells were stained with CellMask™ Green Actin Tracking Stain stock solution (Thermo Fisher Scientific). Image acquisition and analyses were carried out using a Nikon A1 confocal microscope (Tokyo, Japan) and NIS-Elements Ar image analysis (Nikon), respectively.

### 4.6. Analysis of Transcription Factors

After 1 h of treatment with ASD, with or without pretreatment with KRG or Rg3 for 1 h, nasal epithelial cells were harvested and lysed in an ice-cold lysis buffer (Thermo Fisher Scientific). Whole cell lysates were collected and subjected to sodium dodecyl sulfate polyacrylamide gel electrophoresis to separate proteins, and were then transferred onto a nitrocellulose membrane (Bio-Rad). The membranes were blocked with 5% skim milk solution and were incubated with antibodies against phosphorylated ERK, phosphorylated JNK, phosphorylated p38, phosphorylated-Akt, phosphorylated-mTOR, and GAPDH (Santa Cruz Biotechnology). After 1 h of incubation, the membranes were washed with Tris-buffered saline with 0.1% Tween 20 and were then treated with peroxidase-conjugated anti-rabbit immunoglobulin G (Santa Cruz Biotechnology). Bands were visualized using horseradish peroxidase-conjugated secondary antibodies and an enhanced chemiluminescence system (Pierce). Band densities were measured using the multi Gauge v.2.02 software (Fujifilm) and were expressed as a percentage of treated versus untreated cells.

### 4.7. Cell Migration Scratch Assays

Wound scratch assay was conducted as previously described [7]. Nasal epithelial cells were grown in 6-well tissue culture plates until they reached 80% confluence. A straight scratch was made using a pipette tip to create a rupture. The scratched cells were washed twice to remove floating cells, and were then incubated with 50 μg/mL of ASD with or without various concentrations of KRG or Rg3 for 48 h. The cells were photographed to determine the closure of wound that crossed into the scratch areas as compared to zero time.

### 4.8. Statistical Analysis

All experiments were performed with at least five independent individuals and every experiment was performed in duplicate. They produced comparable results. Results are presented as mean ± standard deviation. Statistical significance of the cytotoxic effects of ASD, KRG, and Rg3 was determined using single-factor repeated measure analysis. Student’s t-test was used for comparisons between two groups, while data comparisons among several groups were made using one-way analysis of variance followed by Turkey’s test (SPSS ver. 21.0; IBM Corp., Armonk, NY, USA). A *p*-value of 0.05 or less was considered statistically significant.

## 5. Conclusions

ASD induced EMT by transforming nasal epithelial cells into the fibronectin mesenchymal marker, and downregulating the E-cadherin epithelial marker, through the activation of the transcription factors ERK, p38, and mTOR. KRG and Rg3 inhibited the ASD-induced activation of the ERK, p38, and mTOR signaling pathways, resulting in the up-regulation of E-cadherin, downregulation of fibronectin, and suppressed migration of epithelial cells through the inhibition of F-actin expression. These results indicate the potential role of KRG and Rg3 as therapeutic agents for ASD- or PM-induced nasal mucosal remodeling in CRS.

## Figures and Tables

**Figure 1 molecules-27-02642-f001:**
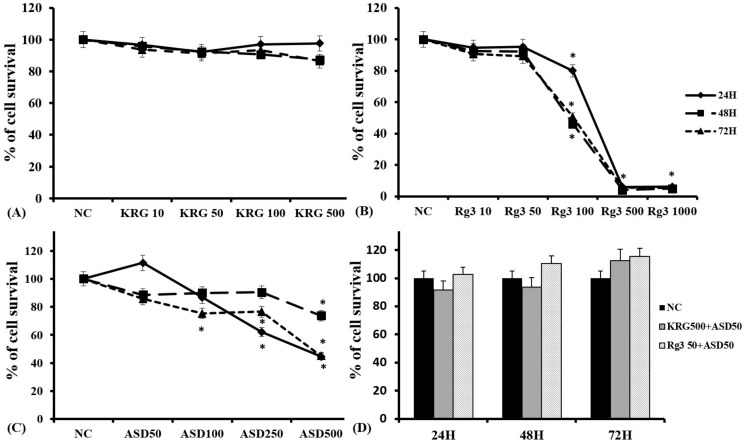
Effects of Korean red ginseng (KRG) and ginsenoside Rg3, and Asian sand dust (ASD), on nasal epithelial cells at various concentrations and times using the CellTiter-96^®^ aqueous cell proliferation assay. Cell survival significantly decreased at 100 μg/mL of Rg3 (**B**) and 100 μg/mL ASD (**C**). However, less than 500 μg/mL of KRG (**A**) or 50 μg/mL of ASD, and 500 μg/mL of KRG or 50 μg/mL of Rg3 (**D**), did not affect the survival of nasal epithelial cells. *: *p* < 0.05 compared with negative control (NC); NC means nasal epithelial cells were not treated with ASD, KRG, or Rg3, *n* = 5.

**Figure 2 molecules-27-02642-f002:**
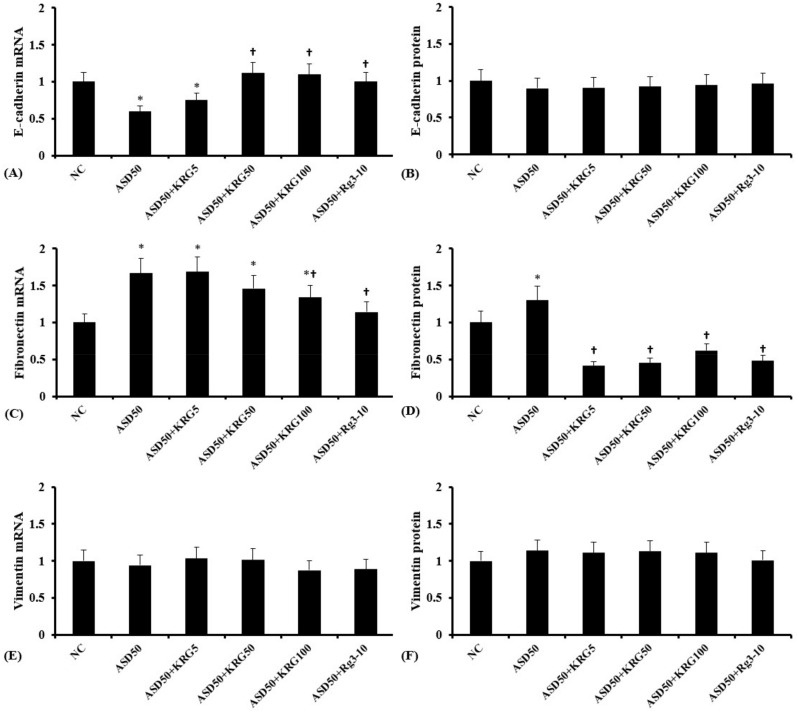
Effects of Korean red ginseng (KRG) and ginsenoside Rg3 on Asian sand dust (ASD)-induced EMT marker mRNA and protein alterations in nasal epithelial cells. Cells were pretreated with or without KRG (5, 50, or 100 μg/mL) or Rg3 (10 μg/mL), and were then stimulated with ASD (50 μg/mL). KRG and Rg3 suppressed the ASD-induced downregulation of E-cadherin mRNA (**A**) expression and the upregulation of fibronectin mRNA (**C**) and protein (**D**) expression in nasal epithelial cells. However, ASD, KRG, and Rg3 did not influence the expression of E-cadherin protein (**B**), vimentin mRNA (**E**), and protein (**F**) expression. NC: negative control (NC means nasal epithelial cells were not treated with ASD, KRG, or Rg3). *: *p* < 0.05 compared with NC. †: *p* < 0.05 compared with the ASD-stimulated group, *n* = 5.

**Figure 3 molecules-27-02642-f003:**
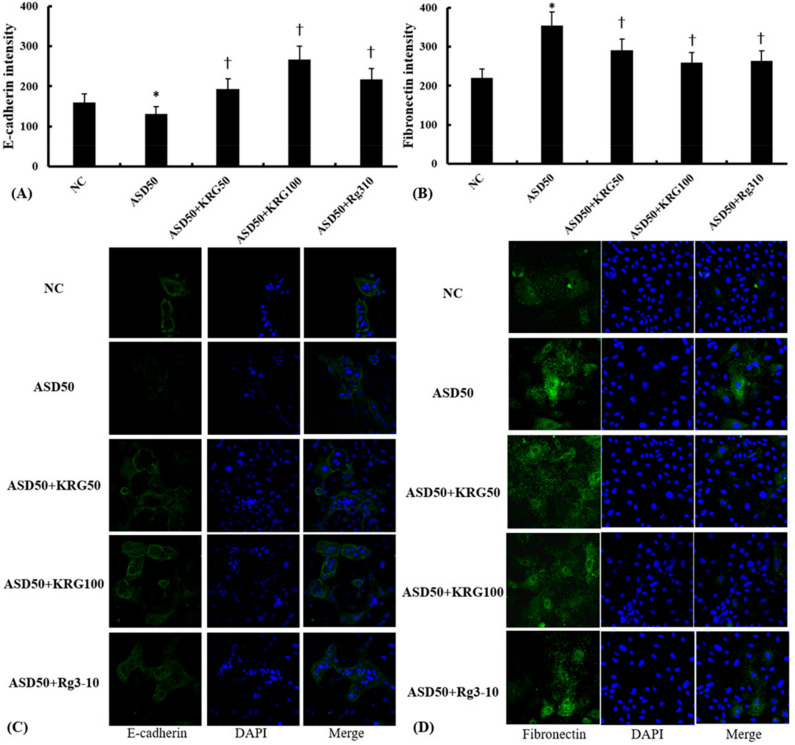
Effects of Korean red ginseng (KRG) and ginsenoside Rg3 on Asian sand dust (ASD)-induced E-cadherin and fibronectin immunofluorescence intensity in nasal epithelial cells. KRG (50 and 100 μg/mL) and Rg3 (10 μg/mL) suppressed the ASD (50 μg/mL)-induced downregulation of E-cadherin (**A**) and the upregulation of fibronectin (**B**) expression in nasal epithelial cells. (**C**,**D**) show representative fluorescein immunofluorescence images with EMT markers (green) and nuclear DAPI (blue). NC: negative control (NC means nasal epithelial cells were not treated with ASD, KRG, or Rg3). *: *p* < 0.05 compared with NC. †: *p* < 0.05 compared with the ASD-stimulated group, *n* = 5.

**Figure 4 molecules-27-02642-f004:**
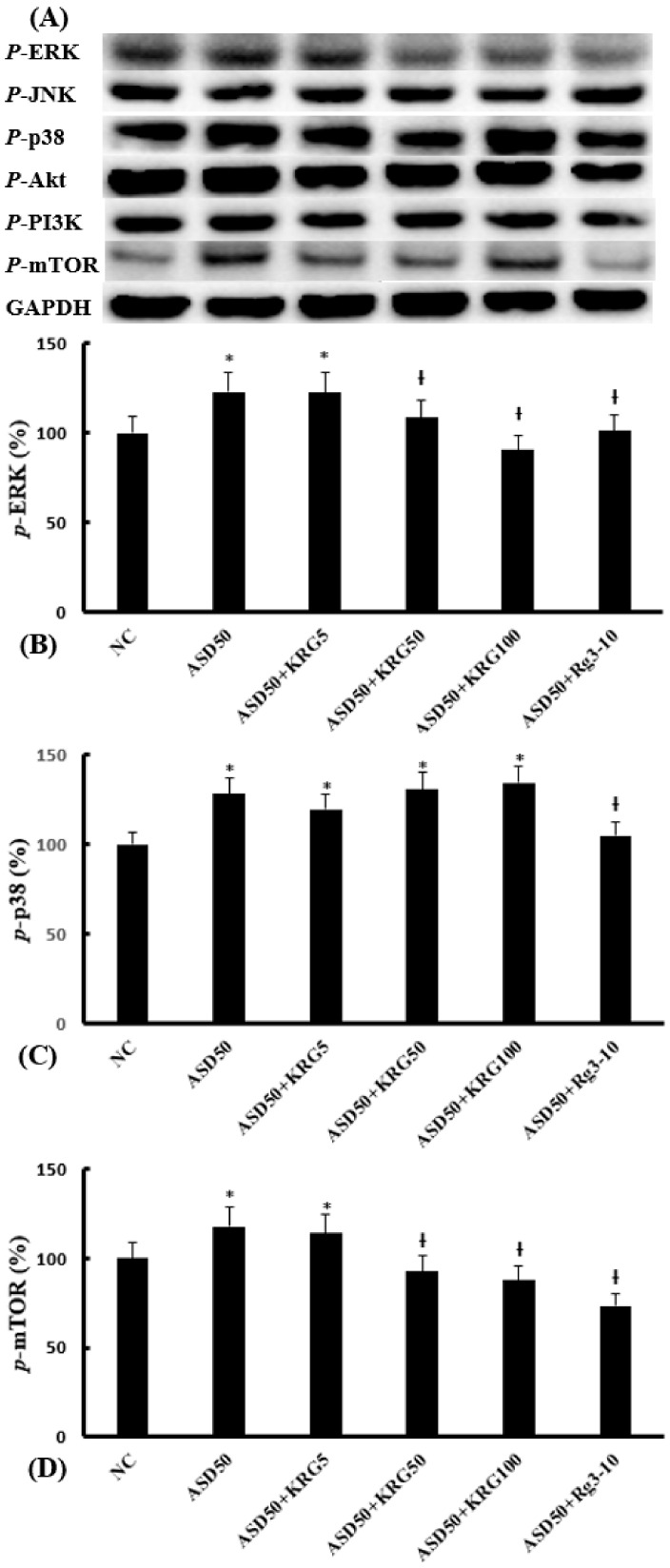
Effects of Korean red ginseng (KRG) and ginsenoside Rg3 on the Asian sand dust (ASD)-induced phosphorylated transcription factor expression in nasal epithelial cells. (**A**) shows representative results of the phosphorylation of transcription factors. ASD (50 μg/mL)-induced phosphorylated-ERK (**B**), phosphorylated-p38 (**C**), and phosphorylated mTOR (**D**) expression was significantly inhibited by KRG (50 and 100 μg/mL) and Rg3 (10 μg/mL). NC: negative control (NC means nasal epithelial cells were not treated with ASD, KRG, or Rg3). *: *p* < 0.05 compared with NC. †: *p* < 0.05 compared with the ASD-stimulated group, *n* = 5.

**Figure 5 molecules-27-02642-f005:**
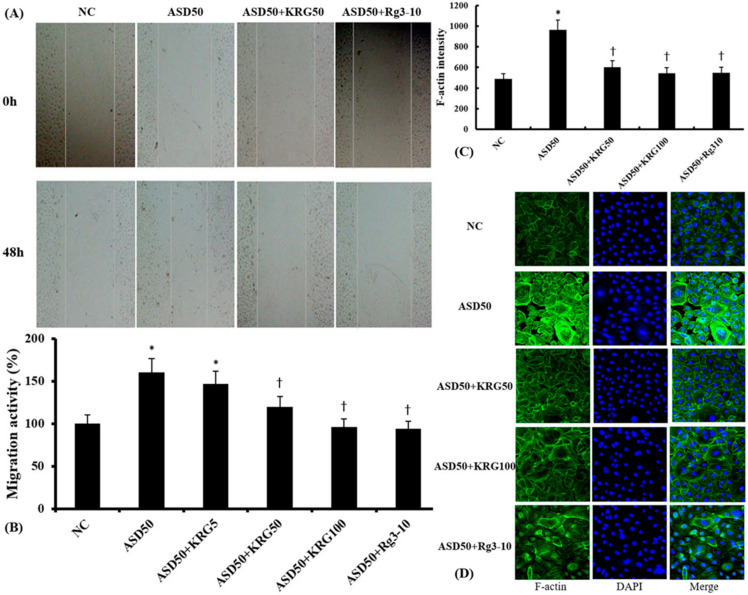
Effects of Korean red ginseng (KRG) and ginsenoside Rg3 on the migration ability and F-actin expression of Asian sand dust (ASD)-stimulated nasal epithelial cells measured using the cell migration assay and immunofluorescence study. (**A**) shows representative images of the cell migration scratch assay. KRG (50 and 100 μg/mL) and Rg3 (10 μg/mL) significantly inhibited ASD-induced cell migration (**B**) and F-actin expression (**C**). (**D**) shows the representative fluorescein immunofluorescence images of F-actin. NC: negative control (NC means nasal epithelial cells were not treated with ASD, KRG, or Rg3). *: *p* < 0.05 compared with NC. †: *p* < 0.05 compared with ASD-stimulated group, *n* = 5.

**Figure 6 molecules-27-02642-f006:**
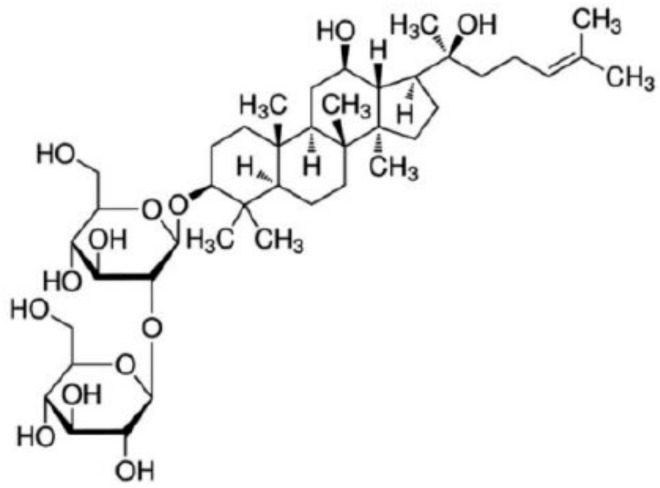
Chemical structure of ginsenoside Rg3.

**Table 1 molecules-27-02642-t001:** The effect of Asian sand dust (ASD), Korean red ginseng (KRG), and ginsenoside Rg3 on IL-8 and TGF-β1 production from nasal epithelial cells.

	IL-8 (pg/mL)	TGF-β1 (pg/mL)
Negative control	180.8 ± 110.5	26.0 ± 3.3
ASD50 (μg/mL)	806.6 ± 386.4 *	46.6 ± 4.7 *
ASD50 + KRG5 (μg/mL)	816.2 ± 302.8 *	32.3 ± 5.0
ASD50 + KRG50 (μg/mL)	842.3 ± 420.7 *	27.8 ± 2.4 ^†^
ASD50 + KRG100 (μg/mL)	897.1 ± 297.6 *	26.5 ± 3.4 ^†^
ASD50 + Rg3-10 (μg/mL)	808.3 ± 415.4 *	27.2 ± 3.7 ^†^

*: *p* < 0.05 compared with negative control, ^†^: *p* < 0.05 compared with the ASD-stimulated group.

**Table 2 molecules-27-02642-t002:** Chemical composition of Asian sand dust particles.

Component	Fraction (%)
SiO_2_	52.13
Al_2_O_3_	15.80
Fe_2_O_3_	5.85
CaO	4.46
K_2_O	2.57
MgO	2.43
Na_2_O	1.59
TiO_2_	0.83
P_2_O_5_	0.18
MnO	0.13
ZnO	0.05
BaO	0.02
SrO	0.02
Other elements	0.22
Loss ignition	13.72

## Data Availability

Data supporting this study can be obtained by contacting the corresponding author.

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
