# Peer review of "Korean Red Ginseng and Ginsenoside Rg3 Suppress Asian Sand Dust-Induced Epithelial–Mesenchymal Transition in Nasal Epithelial Cells"

_molecules, 2022, doi:10.3390/molecules27092642_

Round 1

Reviewer 1 Report

Please explain what is negative control.

In the title you apotrophed Korean Red ginseng. however, in the text it is not enough explain, as well as in the material and method. You say: ...used as traditional herbal medicines to treat various diseases... Please expand... Further ...The commercial standardized water extract of KRG... Please give its chemical composition.

Author Response

I thank the editors and referees of the ‘Molecules’ for taking their time to review my article.

I have made some corrections in the manuscript after going over the referee’s comments.

Please explain what is negative control.

Answer) To clarify the meaning of negative control, ‘NC means, nasal epithelial cells were not treated with ASD, KRG, or Rg3’ in legends of Fig 1 to 5.

In the title you apotrophed Korean Red ginseng. however, in the text it is not enough explain, as well as in the material and method. You say: ...used as traditional herbal medicines to treat various diseases... Please expand... Further ...The commercial standardized water extract of KRG... Please give its chemical composition.

Answer) Korean Red ginseng and their extracts production methods and their contents were mention in Materials and Methods, line 258 to 265 as ‘Fresh ginseng was steamed at 90−100℃, and then ………………………….., Rb 1: 5.8 mg/g, Rb2: 2.3 mg/g, and Rc: 1.7 mg/g. Analysis were performed from general analytic method of ginsenosides from Korea Food and Drug Administration [29]. Chemical structure of Rg3 is proposed in Figure 6.. And chemical structure of Rg2 was described in figure 6.

I hope the revised manuscript will better meet the requirements of the ‘Molecules’ for publication.

Thank you.

Reviewer 2 Report

The authors aimed  to investigate whether epithelial-mesenchymal transition (EMT) can be induced  by stimulation with Asian sand dust (ASD), and to evaluate the effect of Korean Red ginseng (KRG) and ginsenoside Rg3 on ASD-induced EMT in nasal epithelial cells.

The structure of the manuscript appears adequate. The methodology is correctly described with enough experimental data and results to support the work.

This reviewer would like to suggest some improvements and clarifications to achieve the amelioration of the final article. See the comments below:

a. Authors must pay attention to the technical terms acronyms they used in the text.

b. Limitations of the study needs to be added.

Author Response

I thank the editors and referees of the ‘Molecules’ for taking their time to review my article.

I have made some corrections in the manuscript after going over the referee’s comments.

  1. Authors must pay attention to the technical terms acronyms they used in the text.

Answer) To improve the reader's understanding, we added explanations of acronyms in Figs. Added the concentrations of stimulants and clarified the meaning of negative control.

  1. Limitations of the study needs to be added.

Answer) The limitations of this study were mentioned at the last part of Discussion line 242 to 252, as ‘There are some limitations in explaining the immunopharmacologic phenomenon occurring in the sinonasal mucosa. We demonstrated ……………………….the Korea Meteorological Administration. And ASD contains various chemical and microbial compo-nents. In this study, we used autoclaved ASD particles and if we use non-treated ASD, it may more strongly induce EMT in the sinonasal mucosa.’

Reviewer 3 Report

Dear Authors,

After the review process, I have several comments: you should rewrite the abstract section, it contains many general data and include numerical data; you should include references in all Materials and Methods sections; the authors should make comments which could be the role of the nasal microbiota in chronic rhinosinusitis; you should include comments in the Discussion about the bioavailability of functional components from Korean red ginseng; in addition, you should make a correlation between functional compounds bioavailability and inflammatory process similar with the significance of other human microbiota, please see correlations between microbiota bioactivity and bioavailability of functional compounds.

Best regards!

Author Response

I thank the editors and referees of the ‘Molecules’ for taking their time to review my article.

I have made some corrections in the manuscript after going over the referee’s comments.

  1. you should rewrite the abstract section, it contains many general data and include numerical data;

Answer) Results in abstract section, numerical data about TGF-β1 production, e-cadherin and fibronectin expression, and cell migration, were added. In Line 22 to Line 26.

  1. you should include references in all Materials and Methods sections;

Answer) In Materials and Methods, reference 33 to 36 were added in section 4.1., 4.2., 4.5., and 4.7.

  1. the authors should make comments which could be the role of the nasal microbiota in chronic rhinosinusitis; you should include comments in the Discussion about the bioavailability of functional components from Korean red ginseng; in addition, you should make a correlation between functional compounds bioavailability and inflammatory process similar with the significance of other human microbiota, please see correlations between microbiota bioactivity and bioavailability of functional compounds.

Answer) In discussion line229 to 240, we mentioned about microbiota and pharmacokinetics as ‘The pharmakokietic properties KRG or ginsenosides are not completely understood. The absorption rate of orally taken ginseng or ginsenosides is low and their metabolites and biological activity are affected by interaction with gut microbiota, such as Fusobacte-rium, Eubacterium, Bifidobacterium, and Rhodanobacter [29,30]. Topical application of KRG or ginsenosides in the sinonasal mucosa to control local inflammations, may interact with sinonasal microbiota, such as Corynebacterium sp., Staphylococcus aureus, Pseudomonas aeru-ginosa, and Ptptoniphilus, and could change their bioproperties. The active metabolites may disperse and be cleared from the sinonasal mucosa to the gut through mucociliary clearance. Even though active metabolites enter into the gut, only a small portion of metabolites may show bioavailability. Therefore, topical application of KRG or ginsenosides may be more efficient and safer for use in clinical situations. Further studies are needed to understand the bioavailability and pharmacokinetic mechanisms of topically applied ginseng compounds in sinonasal mucosa.

Since, we did not study about the role of microbiota in the sinonasal mucosa and their role in pharmacokinetics. So I cannot answer your comments in detail.

I hope the revised manuscript will better meet the requirements of the ‘Molecules’ for publication.

Thank you.

Round 2

Reviewer 1 Report

After the corrections manuscript "Korean Red ginseng and ginsenoside Rg3 suppress Asian sand dust induced epithelial-mesenchymal transition in nasal epithelial cells" is suitable for publication in Molecules.

Reviewer 3 Report

No additional comments.